# DecompGAIL: Learning Realistic Traffic Behaviors with Decomposed Multi-Agent Generative Adversarial Imitation Learning

**Ke Guo[1], Haochen Liu[1], Xiaojun Wu[2], Chen Lv[1†]**

[1] School of Mechanical and Aerospace Engineering, Nanyang Technological University
[2] Desay SV Automotive
`{ke.guo, lyuchen}@ntu.edu.sg, haochen002@e.ntu.edu.sg, Xiaojun.Wu@desaysv.com`
[†] Corresponding author

## Abstract

Realistic traffic simulation is critical for the development of autonomous driving systems and urban mobility planning, yet existing imitation learning approaches often fail to model realistic traffic behaviors. Behavior cloning suffers from covariate shift, while Generative Adversarial Imitation Learning (GAIL) is notoriously unstable in multi-agent settings. We identify a key source of this instability—irrelevant interaction misguidance—where a discriminator penalizes an ego vehicle's realistic behavior due to unrealistic interactions among its neighbors. To address this, we propose Decomposed Multi-agent GAIL (DecompGAIL), which explicitly decomposes realism into ego–map and ego–neighbor components, filtering out misleading neighbor–neighbor and neighbor–map interactions. We further introduce a social PPO objective that augments ego rewards with distance-weighted neighborhood rewards, encouraging overall realism across agents. Integrated into a lightweight SMART-based backbone, DecompGAIL achieves state-of-the-art performance on the WOMD Sim Agents 2025 benchmark.

## 1 Introduction

Realistic traffic simulation is a cornerstone for intelligent transportation systems, autonomous driving, and urban mobility planning. High-fidelity simulators enable safe, controlled evaluation of driving policies, stress-testing of safety-critical scenarios, and principled analysis of interventions (Wilkie et al., 2015; Li et al., 2017; Pan et al., 2025). Yet achieving realism remains challenging due to the diversity and complexity of human driving behaviors. Prior rule-based models (Treiber et al., 2000; Zhang et al., 2021b), while interpretable and efficient, struggle to capture the complexity and diversity of real-world driving. Behavior cloning (BC) (Michie et al., 1990)—as used in Bergamini et al. (2021); Xu et al. (2023); Philion et al. (2024)—casts imitation as supervised learning from expert trajectories, but suffers from *covariate shift* (Ross et al., 2011): the state distribution induced by a learned policy drifts from that of the expert, compounding errors over time.

These limitations have motivated inverse reinforcement learning (IRL) (Ziebart et al., 2008), which learns a reward function that explains expert behavior and then trains a policy via reinforcement learning (RL). By aligning expert and learner trajectory distribution (Ghasemipour et al., 2019), IRL theoretically addresses *covariate shift*. Among IRL approaches, GAIL (Ho & Ermon, 2016) has gained popularity for traffic modeling (Bhattacharyya et al., 2018; Behbahani et al., 2019; Bhattacharyya et al., 2019; Zheng et al., 2020; Wei et al., 2021; Koeberle et al., 2022; Xin et al., 2024), as it bypasses explicit reward inference by training a discriminator to provide a surrogate reward that encourages policies to match the expert's trajectory distribution. However, in multi-agent settings GAIL often suffers from training instability. To mitigate this, prior work (Bhattacharyya et al., 2018) introduces parameter sharing for the policy and discriminator, and curriculum learning strategies that gradually increase the number of vehicles (Song et al., 2018) or the rollout horizon (Behbahani et al., 2019). While these techniques provide partial improvements, they do not target the root cause and therefore fail to fully resolve the instability.

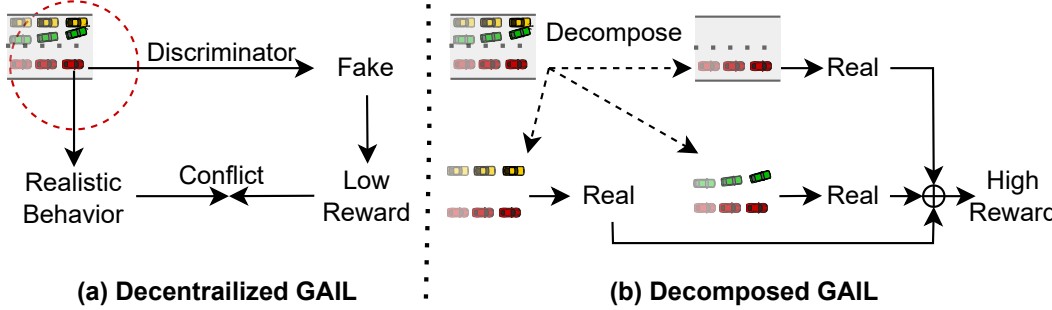

Figure 1: **Comparison with standard decentralized GAIL.** (a) A standard decentralized discriminator evaluating local observations can yield spuriously low rewards for ego-realistic behavior due to unrealistic neighbor–neighbor interactions. (b) **DecompGAIL** separately assesses ego–map and ego–neighbor realism, combining them to obtain a high reward for expert-like ego behavior even when neighbors misbehave.

To investigate this instability, consider the scenario in fig. 1. An ego (red) vehicle, controlled by the learned policy, drives realistically. A nearby (green) vehicle, also policy-controlled, behaves unrealistically and collides with another (yellow) vehicle. A standard decentralized discriminator evaluating the ego's local observation will output a low realism score, because expert data rarely contains such neighbor–neighbor collisions. This yields a contradiction: realistic ego behavior is penalized. We hypothesize that the discriminator overuses weakly relevant neighbor–neighbor and neighbor–map interactions, which leads to training instability. We term this the **irrelevant interaction misguidance** problem, whose severity grows with the number of input neighbors (as confirmed in fig. 3).

To address **irrelevant interaction misguidance**, we propose *Decomposed Multi-agent Generative Adversarial Imitation Learning* (**DecompGAIL**). For each agent, we explicitly disentangle realism into (i) ego–map realism and (ii) ego–neighbor realism by separately discriminating the ego with the static map context and with each neighbor. This input design prevents the discriminator from inferring neighbor–neighbor and neighbor–map interactions—signals that are only weakly correlated with the ego's action yet are influenced by learned policy—yielding rewards that are both more stable and more informative for policy updates. In practice, we further introduce a *social PPO* objective that augments ego rewards with distance-weighted neighborhood rewards, encouraging agents to improve their own realism without degrading that of nearby agents.

This paper makes the following key contributions:

- We identify **irrelevant interaction misguidance**, showing how standard decentralized GAIL discriminators induce instability by overemphasizing weakly relevant neighbor–neighbor interactions.
- We propose **DecompGAIL**, using a decomposed discriminator that separately models ego–scene and ego–neighbor realism while suppressing irrelevant higher-order interactions to yield stable and informative rewards.
- On the WOMD Sim Agents 2025 benchmark, **DecompGAIL** improves training stability over standard decentralized GAIL and achieves state-of-the-art realism.

## 2 RELATED WORK

**Realistic Traffic Modeling.** Traffic modeling can be naturally framed as a multi-agent imitation learning (IL) task (Brown et al., 2020), approached via BC or IRL. BC learns an action distribution conditioned on expert states, parameterized deterministically (Bergamini et al., 2021; Kamenev et al., 2022), as a Gaussian (Yan et al., 2023; Guo et al., 2024; Zhou et al., 2024; Lin et al., 2025), or as a categorical distribution (Philion et al., 2024; Wu et al., 2024; Hu et al., 2024; Zhao et al., 2024). However, BC suffers from *covariate shift* Guo et al. (2023). A direct remedy is data augmentation. Offline approaches perturb demonstrations by injecting noise or dropping trajectory segments (Philion et al., 2024; Wu et al., 2024; Hu et al., 2024). More recent online approaches, such as CAT-K (Zhang

et al., 2025a) and uniMM (Lin et al., 2025), generate augmented trajectories during learning. CAT-K always selects one of the top-$K$ most likely policy actions that moves the agent closest to the expert next state, while uniMM extends this idea to Gaussian mixture policies by choosing the anchor nearest to the expert future trajectory. Nevertheless, these approaches still suffer from distribution shift, since augmentation remains anchored to expert data rather than policy rollouts.

IRL provides a theoretical solution by letting the agent interact with the environment during training. Among IRL methods, GAIL (Ho & Ermon, 2016) is most widely used in driving behavior modeling (Kuefler et al., 2017), but performance degrades in multi-agent domains because of the nonstationary training environment. To stabilize training, prior works (Bhattacharyya et al., 2018; Behbahani et al., 2019; Bhattacharyya et al., 2019; Zheng et al., 2020; Wei et al., 2021; Koeberle et al., 2022; Xin et al., 2024) often employ parameter sharing, where all agents use the same policy and discriminator network, thereby pooling experience across agents. Curriculum learning is also applied, gradually increasing the number of vehicles (Song et al., 2018; Chen et al., 2022) or the rollout horizon (Behbahani et al., 2019). However, these methods still produce undesirable behaviors such as off-map driving, collisions, and abrupt braking (Bhattacharyya et al., 2019). To mitigate this, some works add hand-crafted rewards alongside adversarial rewards (Bhattacharyya et al., 2019; Xin et al., 2024; Bhattacharyya et al., 2022), or purely engineered objectives (Peng et al., 2024; Chen et al., 2025). Yet such hand-crafted rewards introduce prior bias and may fail to capture the complexity of human driving behavior.

**Multi-agent Generative Adversarial Imitation Learning.** Beyond traffic modeling, multi-agent GAIL has been explored in other domains. However, most approaches (Song et al., 2018; Yu et al., 2019; Zhang et al., 2021a) are evaluated in small-scale settings with few agents, where spurious cross-agent interactions are weak. Multi-agent GAIL (Song et al., 2018) employs a centralized discriminator that produces a single shared reward, complicating credit assignment and hindering scalability. MA-DAAC (Jeon et al., 2020) combines decentralized actor–critic training with adversarial learning, improving stability in cooperative tasks but still struggling to disentangle individual contributions. BM3IL (Yang et al., 2020) enhances scalability via a mean-field approximation that averages other agents' actions as discriminator input, which reduces variance but limits fine-grained interaction modeling. Other works add regularization (e.g., Wasserstein objectives (Arjovsky et al., 2017)) to curb discriminator overfitting, at the cost of expressiveness. To our knowledge, prior work has not directly analyzed the root cause of instability in multi-agent GAIL.

## 3 BACKGROUND

**Markov Game.** A Markov game (Littman, 1994) is an extension of the Markov Decision Process (MDP) to multiple agents. A Markov game with $N$ agents is defined by the tuple $\langle \mathcal{S}, \{\mathcal{A}_i\}_{i=1}^N, P, \{r_i\}_{i=1}^N, \gamma \rangle$, where $\mathcal{S}$ is the state space, $\mathcal{A}_i$ is the action space of agent $i$, $P : \mathcal{S} \times \mathcal{A}_1 \times \cdots \times \mathcal{A}_N \to \Delta(\mathcal{S})$ is the transition function, $r_i : \mathcal{S} \times \mathcal{A}_1 \times \cdots \times \mathcal{A}_N \to \mathbb{R}$ is the reward function for agent $i$, and $\gamma \in [0, 1)$ is the discount factor. Each agent $i$ is equipped with a policy $\pi_i : \mathcal{S} \to \Delta(\mathcal{A}_i)$, which maps states to distributions over actions. For notational convenience, we use bold symbols to denote joint quantities; for example, $\boldsymbol{\pi}$ denotes the joint policy, $\boldsymbol{a}$ the joint action.

**Multi-agent Generative Adversarial Imitation Learning.** The objective of multi-agent IL is to learn a joint policy $\boldsymbol{\pi}_\theta$ that mimics the expert joint policy $\boldsymbol{\pi}_E$, given a set of expert demonstrations $\mathcal{D}_E = \{\tau_1, \ldots, \tau_D\}$ with trajectories $\tau = (s_1, \boldsymbol{a}_1, \ldots, s_T, \boldsymbol{a}_T)$. GAIL casts IL as a two-player minimax game between a policy and a discriminator: the discriminator $D_\phi$ is trained to distinguish expert trajectories from policy rollouts, while the policy is optimized to produce trajectories that fool the discriminator. Multi-agent GAIL extends this framework to multiple agents, using either a centralized discriminator (which outputs a shared reward for all agents using all agents' observations) or decentralized discriminators (one per agent, each evaluating its own observations and actions). In this work, we focus on the decentralized setting due to its alignment with human driving decision-making. The objective is:

$$\mathcal{L}_{GAIL} = \min_\theta \max_\phi \mathbb{E}_{\boldsymbol{\pi}_E}\left[\sum_{i=1}^N \log D_\phi(s, a_i)\right] + \mathbb{E}_{\boldsymbol{\pi}_\theta}\left[\sum_{i=1}^N \log\left(1 - D_\phi(s, a_i)\right)\right]. \quad (1)$$

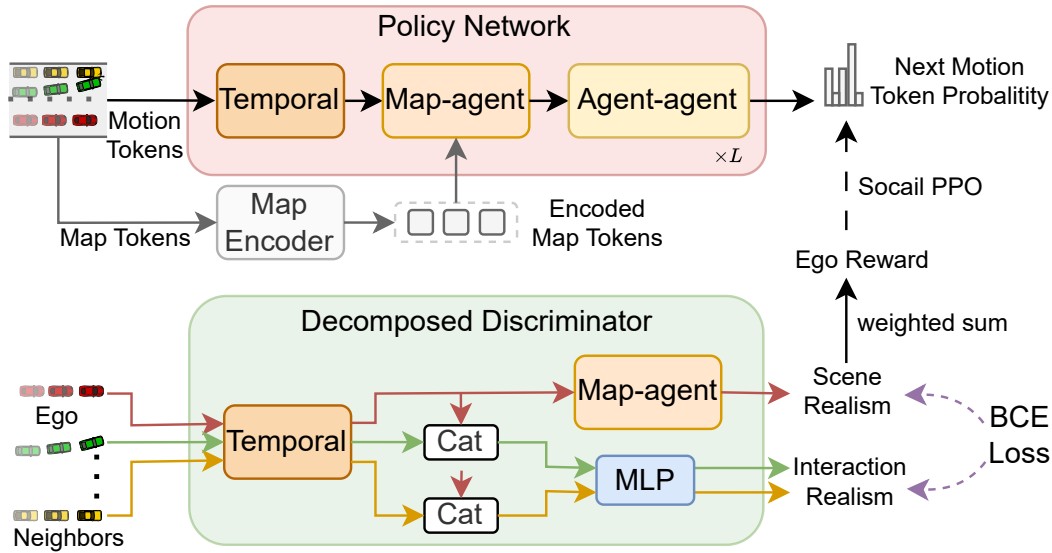

Figure 2: **Overview of the DecompGAIL framework** with three components: a *Map Encoder* (gray) extracting map features; a *Policy Network* (red) predicting motion-token distributions; and a *Decomposed Discriminator* (green) separately assessing scene (ego–map) and interaction (ego–neighbor) realism for expert and policy trajectories. A weighted combination forms each agent's reward, which is then augmented with neighborhood rewards to build the social reward used by PPO training.

Training alternates between updating the discriminator $D_\phi$ with a binary cross-entropy (BCE) loss to classify expert versus policy samples, and updating the joint policy $\pi$ via multi-agent RL using the surrogate reward $r_i = -\log(1 - D_\phi(s, a_i))$.

## 4 APPROACH

The overall framework of **DecompGAIL** is shown in fig. 2. We first pretrain the map encoder and policy network with BC to provide a strong initialization and reduce compute. We then fine-tune the policy with **DecompGAIL**: a decomposed discriminator separately evaluates scene (ego–map) and interaction (ego–neighbor) realism, explicitly omitting neighbor–neighbor and neighbor–map terms, and is trained with a BCE objective using expert and policy samples. The ego reward is the weighted sum of the scene score and all pairwise interaction scores. PPO then optimizes a social reward that augments the ego reward with weighted neighbor rewards. The following sections detail **DecompGAIL**'s implementation within a widely used, scalable Transformer architecture.

### 4.1 POLICY PRETRAINING

**Policy Network.** We instantiate our framework within SMART (Wu et al., 2024), a widely used learning-based traffic model adopted by prior work (Zhang et al., 2025a;b; Ahmadi & Schofield, 2025). SMART parameterizes the joint action as independent categorical distributions over a shared vocabulary of motion tokens. At each time step, it predicts every agent's next-step token distribution conditioned on encoded map tokens $m$ and all agents' past motion tokens $\boldsymbol{a}_{<t} := (\boldsymbol{a}_0, \ldots, \boldsymbol{a}_{t-1})$:

$$\boldsymbol{\pi}_\theta(\boldsymbol{a}_t \mid s_t) = \prod_{i=1}^{N} \pi_\theta(a_t^i \mid \boldsymbol{a}_{<t}, m), \qquad (2)$$

where all agents share a single policy $\pi_\theta$.

Specifically, SMART employs a factorized Transformer (Ngiam et al., 2021) with multi-head self-attention (MHSA) and multi-head cross-attention (MHCA) to encode map–motion interactions. The map encoder first produces encoded map tokens $m$ by attending over spatial neighboring map tokens with MHSA. Then, for each agent $i$ at time $t$, the motion encoder takes the current motion-token

embedding $e_t^i$ as the query and sequentially applies temporal, map–agent, and agent–agent attention layers:

$$\text{temp}_t^i = \text{MHSA}\big(q(e_t^i),\ k(e_{t-\tau}^i, \text{RPE}_{t,t-\tau}^i),\ v(e_{t-\tau}^i, \text{RPE}_{t,t-\tau}^i)\big), \quad 0 < \tau < t \tag{3a}$$

$$\text{map}_t^i = \text{MHCA}\big(q(\text{temp}_t^i),\ k(m^j, \text{RPE}_t^{ij}),\ v(m^j, \text{RPE}_t^{ij})\big), \quad j \in \mathcal{N}_i \tag{3b}$$

$$\text{agent}_t^i = \text{MHSA}\big(q(\text{map}_t^i),\ k(\text{map}_t^j, \text{RPE}_t^{ij}),\ v(\text{map}_t^j, \text{RPE}_t^{ij})\big), \quad j \in \mathcal{N}_i \tag{3c}$$

where RPE denotes relative positional encoding (Cui et al., 2022), and $\mathcal{N}_i$ is the neighborhood set of map or agent tokens determined by a distance threshold. After $L$ stacked layers of temporal, map–agent, and agent–agent attentions, an MLP head outputs the next-token distribution for each agent.

**BC Pretraining.** We initialize the policy network by BC to accelerate training, since online interaction in GAIL is more computationally expensive. BC maximizes the likelihood of the expert joint action at each time step, conditioned on the expert's past motions and map tokens:

$$\mathcal{L}_{BC} = \max_\theta \mathbb{E}_{\boldsymbol{\pi}_E}[\log \pi_\theta(\boldsymbol{a}_t \mid \boldsymbol{a}_{<t}, m)], \tag{4}$$

This pretraining step enables the model to imitate expert behavior on observed trajectories. However, during testing rollouts, small prediction errors can accumulate, leading agents into states not encountered in the demonstrations (Ross et al., 2011). This distributional shift often results in implausible behaviors such as collisions or off-road driving (Tian & Goel, 2025).

## 4.2 DECOMPOSED GAIL FINE-TUNING

**Parameter-sharing GAIL.** To address the *covariate shift* issue, we apply GAIL to online fine-tune the BC-pretrained policy. A widely adopted practice in prior works (Bhattacharyya et al., 2018; Behbahani et al., 2019; Bhattacharyya et al., 2019) is *parameter-sharing GAIL* (PS-GAIL), which employs a single discriminator shared across all agents. At each time step, the discriminator evaluates every agent's local observation, and the resulting score is used as the reward for updating the shared policy $\pi_\theta$ via single-agent RL. To implement this, one can replace the policy's action head with a binary classification head; the discriminator score for each agent $i$ at time $t$ is:

$$D_\phi(s_t, a_i) = D_\phi(\boldsymbol{a}_{\leq t}^i, \boldsymbol{a}_{\leq t}^{\mathcal{N}_i}, m) = \text{MLP}(\text{agent}_t^i), \tag{5}$$

where $\boldsymbol{a}_{\leq t}^{\mathcal{N}_i} := \{\boldsymbol{a}_{\leq t}^j \mid j \in \mathcal{N}_i\}$ denotes the motion token histories of all $i$'s neighbors. When computing the reward for each agent $i$, we treat that agent as the ego agent.

Despite its simplicity, PS-GAIL is often unstable (Bhattacharyya et al., 2018), degrading simulation quality (see fig. 3). To examine the source, we conceptually decompose the discriminator's signal into four terms:

$$D_\phi^i\big(\boldsymbol{a}_{\leq t}^i, \boldsymbol{a}_{\leq t}^{\mathcal{N}_i}, m\big) = \underbrace{\phi_1(\boldsymbol{a}_{\leq t}^i, m)}_{\text{ego–map (scene) realism}} + \underbrace{\sum_{j \in \mathcal{N}_i} \phi_2(\boldsymbol{a}_{\leq t}^i, \boldsymbol{a}_{\leq t}^j)}_{\text{ego–neighbor (interaction) realism}}$$
$$+ \underbrace{\phi_3(\boldsymbol{a}_{\leq t}^{\mathcal{N}_i}, m)}_{\text{neighbor–map / neighbor–neighbor}} + \underbrace{\phi_4(\boldsymbol{a}_{\leq t}^i, \boldsymbol{a}_{\leq t}^{\mathcal{N}_i}, m)}_{\text{higher order}}. \tag{6}$$

Here, $\phi_1$ measures alignment of the ego trajectory with the map (*scene realism*); $\phi_2$ aggregates ego–neighbor plausibility (*interaction realism*); $\phi_3$ captures neighbor–map and neighbor–neighbor signals; and $\phi_4$ models higher-order correlations among ego, neighbors, and the map. Notably, $\phi_3$ is only weakly coupled to the ego's action, yet its interaction count grows roughly quadratically with neighborhood size, making the overall reward increasingly noisy as $|\mathcal{N}_i|$ increases and thus less informative for ego policy updates. We refer to this phenomenon as **irrelevant interaction misguidance**.

---

**Algorithm 1:** Decomposed GAIL

---

**Input:** Markov game $\mathcal{M}$, expert trajectories $\mathcal{D}_E$, policy $\pi_\theta$, decomposed discriminator $D_{\phi_1,\phi_2}$, value functions $V_\psi$, batch size $B$, discount $\gamma$, GAE $\lambda$, PPO clip $\epsilon$, weights $w_{ij}, \lambda_{ij}$.
**Output:** Learned policy $\pi_\theta$.

---

**while** *not converged* **do**

    Collect $B$ trajectories with $\pi_\theta$ in $\mathcal{M}$;

    **foreach** *agent $i$ and step $t$* **do**

        Compute realism: $S_t^i = \phi_1(\boldsymbol{a}_{\leq t}^i, m)$, $I_t^{ij} = \phi_2(\boldsymbol{a}_{\leq t}^i, \boldsymbol{a}_{\leq t}^j)$, $j \in \mathcal{N}_i$;

    Update discriminator $D_{\phi_1,\phi_2}$ by minimizing

$$\mathcal{L}_D = \mathbb{E}_{\boldsymbol{\pi}_E}\left[\log S_t^i + \sum_j w_{ij} \log I_t^{ij}\right] + \mathbb{E}_{\pi_\theta}\left[\log(1-S_t^i) + \sum_j w_{ij} \log(1-I_t^{ij})\right].$$

    Compute agent reward

$$r_t^i = -\log(1-S_t^i) - \sum_{j\in\mathcal{N}_i} w_{ij} \log(1-I_t^{ij}),$$

    and social reward $r_t^{S_i} = r_t^i + \sum_{j\in\mathcal{N}_i} \lambda_{ij} r_t^j$;

    Estimate advantages $A_t^{S_i}$ and targets $G_t^{S_i}$ with GAE;

    Update $\pi_\theta, V_\psi$ using PPO with clipped surrogate and value loss and BC loss;

---

**Decomposed Discriminator.** This **irrelevant interaction misguidance** arises because the PS-GAIL network in eq. (5) can implicitly represent all four terms: map and agent tokens (positions and features) are fused into a single ego feature before scoring, entangling signals. To avoid this, we replace the monolithic discriminator with a *decomposed* architecture that (i) computes *scene realism* from ego–map inputs and (ii) computes *interaction realism* from pairwise ego–neighbor inputs, thereby suppressing neighbor–neighbor and higher-order signals by design.

Concretely, scene realism uses the temporal and map–agent attention features as in eqs. (3a) and (3b), followed by an MLP:

$$S_t^i = \phi_1(\boldsymbol{a}_{\leq t}^i, m) = \mathrm{MLP}(\mathrm{map}_t^i). \tag{7}$$

Interaction realism is computed per ego–neighbor pair via another MLP over concatenated temporal features:

$$I_t^{ij} = \phi_2(\boldsymbol{a}_{\leq t}^i, \boldsymbol{a}_{\leq t}^j) = \mathrm{MLP}([\mathrm{temp}_t^i, \mathrm{RPE}_t^{ij}, \mathrm{temp}_t^j]), \tag{8}$$

where $[\cdot]$ denotes concatenation. The discriminator loss is

$$\mathcal{L}_D = \mathbb{E}_{\boldsymbol{\pi}_E}\left[\log S_t^i + \sum_{j\in\mathcal{N}_i} w_{ij} \log I_t^{ij}\right] + \mathbb{E}_{\pi_\theta}\left[\log(1 - S_t^i) + \sum_{j\in\mathcal{N}_i} w_{ij} \log(1 - I_t^{ij})\right], \tag{9}$$

with distance-decayed weights $w_{ij} = \alpha \exp(-d(i,j)/\beta)$, where $d$ is inter-agent distance and $\alpha, \beta$ are hyperparameters, to emphasize nearby interactions that are more causally tied to the ego's action. The per-agent reward is

$$r_t^i = -\log(1 - S_t^i) - \sum_{j\in\mathcal{N}_i} w_{ij} \log(1 - I_t^{ij}). \tag{10}$$

**Social Policy Learning.** A straightforward way to optimize a decentralized multi-agent policy is independent PPO (IPPO) (de Witt et al., 2020), which treats each agent as if in a single-agent environment. However, optimizing purely individual objectives can degrade the overall realism of the population (Schwarting et al., 2019). We therefore define a social reward for agent $i$:

$$r_t^{S_i} = r_t^i + r_t^{\mathcal{N}_i} = r_t^i + \sum_{j\in\mathcal{N}_i} \lambda_{ij} r_t^j \tag{11}$$

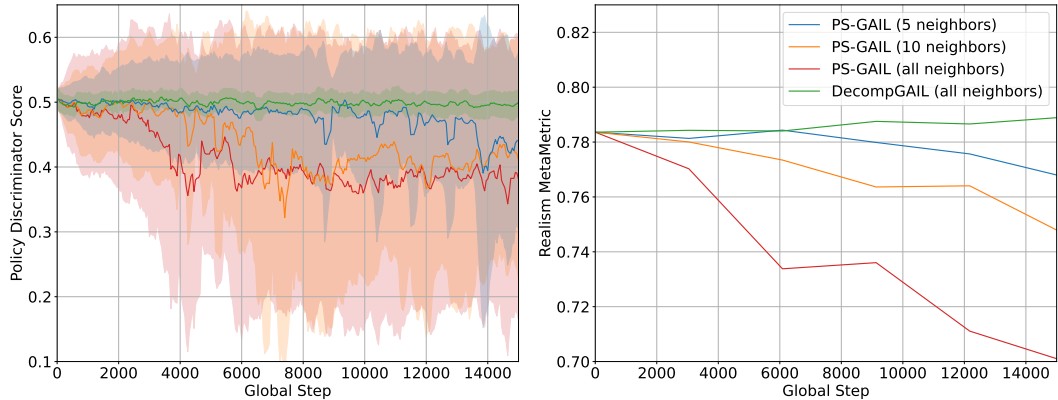

Figure 3: **Training stability.** In the left plot, solid lines denote the mean output realism scores, while shaded areas indicate the standard deviation. The realism meta-metric is evaluated on the 2% validation split. The **DecompGAIL** maintains lower variance and better simulation realism performance than PS-GAIL.

where $r_t^{\mathcal{N}_i}$ aggregates neighborhood rewards and $\lambda_{ij}$ is a distance-decayed weight (analogous to $w_{ij}$). Training then follows IPPO, but with individual rewards replaced by social rewards. Advantages are estimated with Generalized Advantage Estimation (GAE), and each agent's value function is given by an MLP applied to the policy network's final agent–agent attention features. To improve stability, we combine the PPO loss and BC loss. The full procedure is summarized in algorithm 1. More details about training can be found in appendix A.

## 5 EXPERIMENTS

This section addresses the following research questions:

- **Q1:** Does **DecompGAIL** improve training stability and performance over the widely used PS-GAIL method?

- **Q2:** How does **DecompGAIL** compare with state-of-the-art baselines in terms of realism?

- **Q3:** What is the contribution of each component to overall performance?

We first describe the experimental setup, including dataset, metrics, and baselines, before presenting detailed analyses for each question.

### 5.1 EXPERIMENTAL SETUPS

**Dataset.** We conduct experiments on the Waymo Open Motion Dataset (WOMD) (Ettinger et al., 2021), a large-scale benchmark for traffic simulation containing 487k training, 44k validation, and 44k test scenes. Each scenario is recorded at 10 Hz, with 1 second of historical context and 8 seconds of future trajectories for up to 128 traffic participants (vehicles, cyclists, pedestrians), along with high-definition maps. Following CAT-K (Zhang et al., 2025a), we evaluate on 2% of the validation split (880 of 44,097 scenarios).

**Metrics.** We adopt the official metrics from the WOMD Sim Agents 2025 Challenge (WOSAC) (Montali et al., 2024):

- **Kinematic:** weighted likelihood of linear speed, linear acceleration, angular speed, and angular acceleration.

- **Interactive:** weighted likelihood of nearest object distance, collision, and time-to-collision.

- **Map-based:** weighted likelihood of road edge distance, off-road, and traffic-light violation.

Table 1: **Results on the WOSAC 2025 leaderboard test split.** (Waymo, 2025)

| Model | Metametric ↑ | Kinematic ↑ | Interactive ↑ | Map-based ↑ | minADE ↓ |
|---|---|---|---|---|---|
| **SMART-tiny-DecompGAIL (ours)** | **0.7864** | 0.4919 | **0.8152** | 0.9176 | 1.4209 |
| SMART-R1 (Pei et al., 2025) | 0.7858 | **0.4944** | 0.8110 | 0.9201 | 1.2885 |
| SMART-tiny-RLFTSim (Ahmadi & Schofield, 2025) | 0.7857 | 0.4927 | 0.8129 | 0.9183 | 1.3252 |
| TrajTok (Zhang et al., 2025b) | 0.7852 | 0.4887 | 0.8116 | **0.9207** | 1.3179 |
| SMART-tiny-CLSFT (Zhang et al., 2025a) | 0.7846 | 0.4931 | 0.8106 | 0.9177 | 1.3065 |
| UniMM (Lin et al., 2025) | 0.7829 | 0.4914 | 0.8089 | 0.9161 | 1.2949 |
| SMART-tiny (Wu et al., 2024) | 0.7814 | 0.4854 | 0.8089 | 0.9153 | 1.3931 |
| LLM2AD (Wang et al., 2025) | 0.7779 | 0.4846 | 0.8048 | 0.9109 | **1.2827** |
| InfGen (Peng et al., 2025) | 0.7731 | 0.4493 | 0.8084 | 0.9127 | 1.4252 |

The *realism meta-metric* is a weighted combination of these measures and serves as the primary evaluation criterion. We also report *minADE*, which is widely used in motion prediction but not included in the realism meta-metric. All metrics are computed from 32 rollouts of 8 seconds at 10 Hz.

## 5.2 STABILITY (Q1)

To assess stability, we compare **DecompGAIL** to a naive PS-GAIL baseline obtained by replacing our decomposed discriminator with a standard local discriminator while keeping all other components unchanged. We track the discriminator scores and the realism meta-metric during training under four settings: PS-GAIL with 5, 10, and all available nearest neighbors within 60 m, and **DecompGAIL** with all neighbors within the same range.

As shown in fig. 3, PS-GAIL exhibits increasing variance and a lower mean score as neighbor count grows, as the model overfits to penalize unrealistic neighbor–neighbor and neighbor–map interactions. Because these interactions are only weakly correlated with the ego's own behavior, the resulting reward signals are unstable and misleading, ultimately degrading simulation performance. In contrast, **DecompGAIL** maintains low variance with a mean near the expected equilibrium (0.5), providing stable, informative rewards that support steady RL improvement. As a result, its simulation performance improves steadily throughout training.

## 5.3 PERFORMANCE (Q2)

To address Q2, we compare **DecompGAIL** with state-of-the-art baselines on the WOSAC leaderboard.

**Baselines.** We compare both tokenized models (Pei et al., 2025; Ahmadi & Schofield, 2025; Zhang et al., 2025b;a; Wu et al., 2024; Peng et al., 2025; Wang et al., 2025) and continuous model (Lin et al., 2025), covering a broad spectrum of learning paradigms:

- Naive BC (Wu et al., 2024; Zhang et al., 2025b; Peng et al., 2025);
- BC with data augmentation (Zhang et al., 2025a; Lin et al., 2025);
- RL finetuning approaches (Wang et al., 2025; Pei et al., 2025; Ahmadi & Schofield, 2025).

Specifically, SMART-R1 (Pei et al., 2025) applies R1-style reinforcement fine-tuning. SMART-tiny-RLFTSim (Ahmadi & Schofield, 2025) directly optimizes the realism meta-metric as a reward for RL fine-tuning the SMART-tiny model (Wu et al., 2024). TrajTok (Zhang et al., 2025b) enhances SMART by introducing a trajectory tokenizer with better coverage, symmetry, and robustness, along with spatial-aware label smoothing. SMART-tiny-CLSFT (Zhang et al., 2025a) augments training via expert-guided online trajectory generation to mitigate *covariate shift*. UniMM (Lin et al., 2025) employs a continuous Gaussian mixture model with expert-guided data augmentation. LLM2AD (Wang et al., 2025) integrates GRPO post-training with test-time search and clustering on a larger SMART backbone equipped with enhanced tokenization and positional encoding. InfGen (Peng et al., 2025) models the entire scene as a sequence of tokens—including agent states, motion vectors, and traffic signals—and uses a Transformer to autoregressively simulate traffic.

**Results.** As reported in table 1, **DecompGAIL** achieves the best realism meta-metric and consistently outperforms all baselines on interactive metrics, while remaining competitive on kinematic and

Table 2: **Ablation study on WOSAC 2% validation split.**

| Model | Metametric ↑ | Kinematic ↑ | Interactive ↑ | Map-based ↑ | Collision Likelihood ↑ |
|---|---|---|---|---|---|
| w/o DecompGAIL | 0.7836 | 0.5077 | 0.8204 | 0.8926 | 0.9667 |
| w/o scene realism | 0.7801 | 0.5053 | 0.8248 | 0.8795 | 0.9794 |
| w/o interact realism | 0.7772 | 0.5043 | 0.8132 | 0.8869 | 0.9573 |
| mean interact realism | 0.7819 | 0.5140 | 0.8153 | 0.8921 | 0.9635 |
| w/o neighborhood reward | 0.7871 | 0.5146 | 0.8258 | 0.8930 | 0.9788 |
| mean neighborhood reward | 0.7882 | 0.5140 | 0.8282 | 0.8935 | 0.9812 |
| **DecompGAIL** | **0.7889** | **0.5148** | **0.8283** | **0.8948** | **0.9837** |

map-based scores. Its relatively higher minADE can be attributed to the fact that our method prioritizes matching feature distributions between expert and policy trajectories, rather than optimizing for distance-based similarity. Overall, these results demonstrate that **DecompGAIL** fine-tuning delivers substantial improvements in the realism of traffic simulation.

## 5.4 ABLATION (Q3)

The ablation study in Table 2 evaluates the contribution of each component of **DecompGAIL**:

**DecompGAIL Fine-tuning.** We begin by evaluating the model prior to the **DecompGAIL** fine-tuning stage. Omitting this stage ("w/o DecompGAIL") reduces the realism meta-metric from 0.7889 to 0.7836 and lowers the collision likelihood from 0.9837 to 0.9667, indicating that a purely BC-pretrained model suffers from *covariate shift* and produces less realistic simulations.

**Realism Decomposition.** Next, we examine the influence of the realism decomposition and the distance-weighted design. Removing *scene realism* causes a marked decline in map-based scores (0.8795 vs. 0.8948) while leaving interaction performance relatively strong. Conversely, excluding *interaction realism* leads to a sharp degradation in both the interactive metric (0.8132 vs. 0.8283) and the collision likelihood (0.9573 vs. 0.9837), underscoring the necessity of explicitly modeling ego–neighbor interactions for realistic multi-agent behavior. Replacing distance-weighted aggregation with uniform averaging partially recovers performance but still lags behind the distance-decayed variant, highlighting the effectiveness of distance-aware weighting.

**Social Reward.** Finally, we evaluate the role of social rewards. Removing neighborhood rewards ("w/o neighborhood reward") slightly decreases both interactive and map-based scores, while replacing distance-weighted aggregation with uniform averaging ("mean neighborhood reward") also degrades performance. Together, these results demonstrate that distance-decayed social rewards are more effective at promoting realism across all agents.

## 5.5 HYPERPARAMETER

In this section, we analyze the effect of the decay parameters $\alpha$ and $\beta$ used for interaction weighting and social rewards on the realism meta-metric. We first conduct a sweep over decay scales $\alpha \in \{2.5, 5, 10, 20\}$ and decay ranges $\beta \in \{1, 2.5, 5, 10\}$ for the interaction weights, with social rewards disabled. The results in the left subplot of fig. 4 show two key observations: (1) performance is more sensitive to the decay range $\beta$ than to the scale $\alpha$; and (2) extremely small or large $\beta$ overemphasizes very close or very distant neighbors, both of which degrade realism.

Next, fixing the interaction weights at their optimal values ($\alpha = 10$, $\beta = 2.5$), we sweep decay scales $\alpha \in \{0.5, 1, 5, 10\}$ and rates $\beta \in \{1, 2.5, 5, 10\}$ for the social reward weighting. We find in the right subplot of fig. 4 that social reward parameters have a weaker influence than interaction weights. However, very large $\alpha$ or $\beta$ make the neighborhood weights nearly uniform, which increases noise in the social reward and slightly reduces realism.

## 5.6 QUALITATIVE RESULTS

We provide qualitative demonstrations of our fine-tuned model in fig. 5, showcasing one rollout in two distinct scenarios. In both cases, the policy generates realistic multi-agent trajectories, maintaining

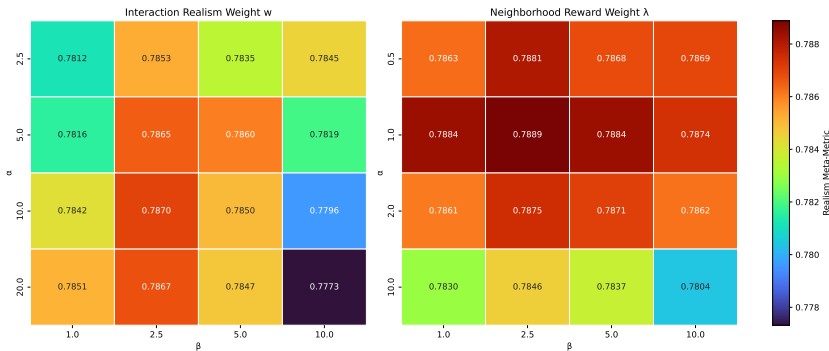

Figure 4: **Influence of decay parameters $\alpha$ and $\beta$ on interaction weighting and social rewards.**

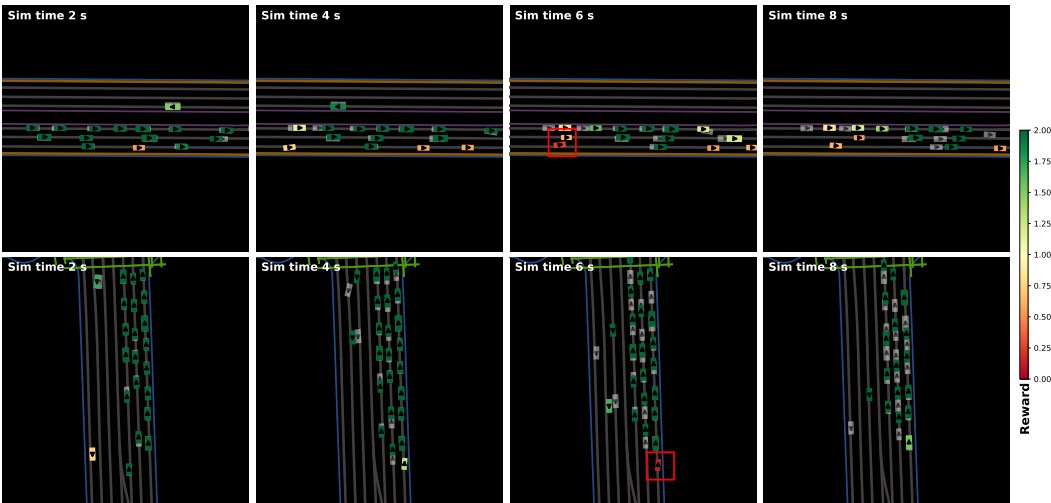

Figure 5: **Qualitative results on WOSAC.** Each row shows a rollout of our model in a different scene. Transparent boxes denote ground-truth agents; solid boxes are agents generated by our model. Agent colors indicate the per-agent reward from our decomposed discriminator. The red rectangle in the first row highlights a low-reward near-collision; the red rectangle in the second row highlights a low-reward off-road tendency.

collision-free and on-road behavior throughout the full 8-second horizon. This highlights the ability of our model to sustain expert-like motion even in complex interactive settings.

To further interpret model behavior, we visualize the per-agent rewards estimated by our decomposed discriminator, encoded as the color of each agent at every timestep. As illustrated, the discriminator consistently assigns lower rewards to agents that deviate from realistic behavior—for example, when a vehicle approaches a near-collision situation or begins drifting toward the road edge. Conversely, agents following smooth and expert-like motion patterns are assigned high rewards. Additional rollout examples, including more challenging interactive scenarios, are provided in the supplementary.

## 6 CONCLUSION

We introduced **DecompGAIL**, a decomposed adversarial imitation learning framework for realistic multi-agent traffic simulation. By explicitly separating *scene* and *interaction* realism in the discriminator, **DecompGAIL** suppresses weakly relevant neighbor–neighbor and neighbor–map signals that otherwise misguide reward learning. Coupled with a *social PPO* objective that augments ego rewards with distance-decayed neighborhood rewards, **DecompGAIL** delivers stable and informative training signals. Empirically, **DecompGAIL** improves training stability over PS-GAIL and achieves state-of-the-art realism on the WOSAC 2025 benchmark.

ACKNOWLEDGMENTS

This work is supported by Nanyang Technological University (NTU)-DESAY SV Research Program under Grant 2018-0980.

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

# A IMPLEMENTATION DETAILS

We pretrain the policy network with BC for 32 epochs following CAT-K (Zhang et al., 2025a), taking about 44 hours, and then fine-tune with DecompGAIL for 2 additional epochs, taking about 10 hours. During fine-tuning, the map encoder is frozen to reduce GPU memory usage and the policy network and discriminator network share the map encoder, enabling larger batch sizes with negligible performance loss. Both pretraining and fine-tuning are conducted on $8 \times$ H800 80GB GPUs with a total batch size of 80. We use Adam with weight decay of 0.01 for optimization, with a policy learning rate of $5 \times 10^{-4}$ during pretraining and $5 \times 10^{-5}$ during fine-tuning, and a discriminator learning rate of $1 \times 10^{-4}$. To stabilize adversarial training, we generate fresh rollouts at each training step using the current policy, then update the discriminator and policy once. Additional hyperparameters for DecompGAIL fine-tuning are provided in table 3.

Table 3: **Hyperparameters for DecompGAIL Fine-tuning.**

| Hyperparameter | Value |
|---|---|
| MLP depth (all) | 2 |
| MLP width (all) | 128 |
| Network dropout (all) | 0 |
| PPO discount $\gamma$ | 0.99 |
| PPO clip $\epsilon$ | 0.2 |
| PPO epochs | 1 |
| PPO batch size | 80 |
| PPO rollout length | 16 |
| GAE $\lambda$ | 0.95 |
| Value loss weight | $1 \times 10^{-3}$ |
| $\alpha$ in $w_{ij}$ | 10 |
| $\beta$ in $w_{ij}$ | 2.5 |
| $\alpha$ in $\lambda_{ij}$ | 1 |
| $\beta$ in $\lambda_{ij}$ | 2.5 |

# B DISCRIMINATOR GRADIENT PENALTY

To evaluate whether common gradient-penalty techniques can mitigate the instability caused by irrelevant interactions, we apply WGAN-GP (Arjovsky et al., 2017) and R1/R2 regularization (Mescheder et al., 2018) (with penalty weight fixed to 1) to both the naive PS-GAIL baseline and our DecompGAIL framework. Results are reported in table 4.

We find that gradient penalties do modestly improve PS-GAIL, largely by constraining the discriminator's expressiveness and thereby reducing the variance of its output scores. However, even with gradient penalties, the PS-GAIL discriminator can still be misled by irrelevant neighbor–neighbor and neighbor–map interactions, so its overall performance remains significantly below that of DecompGAIL. When adding these penalties to DecompGAIL, we observe the opposite trend: gradient penalties reduce overall realism despite slightly improving the kinematic metrics. This is expected—gradient penalties enforce smoothness in the discriminator's mapping, but critical components of interactive and map-based realism such as collision and off-road likelihood are inherently *non-smooth* and depend on sharp decision boundaries. Penalizing discriminator gradients therefore harms the model's ability to detect these events, degrading the interactive and map-based metrics.

Table 4: **Effect of discriminator gradient penalties evaluated on the 2% validation split.**

| Model | Metametric ↑ | Kinematic ↑ | Interactive ↑ | Map-based ↑ | Collision Likelihood ↑ | Offroad Likelihood ↑ |
|---|---|---|---|---|---|---|
| PS-GAIL | 0.7674 | 0.4830 | 0.8012 | 0.8864 | 0.9494 | 0.9464 |
| PS-GAIL + WGAN-GP | 0.7723 | 0.5011 | 0.8111 | 0.8654 | 0.9538 | 0.9322 |
| PS-GAIL + R1 | 0.7776 | 0.5011 | 0.8198 | 0.8814 | 0.9704 | 0.9370 |
| PS-GAIL + R2 | 0.7706 | 0.5037 | 0.8091 | 0.8736 | 0.9484 | 0.9258 |
| DecompGAIL + WGAN-GP | 0.7865 | **0.5186** | 0.8255 | 0.8894 | 0.9796 | 0.9441 |
| DecompGAIL + R1 | 0.7861 | 0.5157 | 0.8233 | 0.8928 | 0.9734 | 0.9493 |
| DecompGAIL + R2 | 0.7840 | 0.5145 | 0.8212 | 0.8900 | 0.9743 | 0.9469 |
| **DecompGAIL** | **0.7889** | 0.5148 | **0.8283** | **0.8948** | **0.9837** | **0.9529** |

## C  UNFREEZING MAP ENCODER

To evaluate the effect of updating the map encoder during **DecompGAIL** training, we conducted an additional experiment in which the map encoder is fine-tuned rather than frozen (see fig. 6). We observe that freezing the map encoder yields slightly better training stability and marginally higher realism.

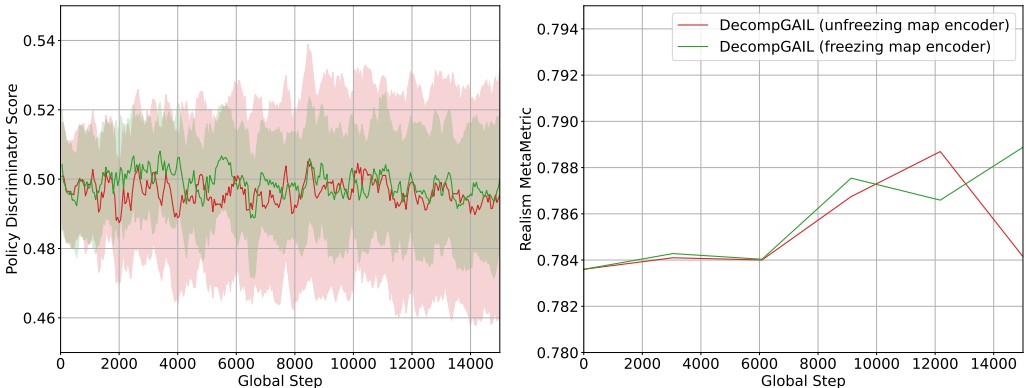

Figure 6: **Effect of freezing vs. fine-tuning the map encoder.** Solid lines show the mean discriminator realism scores, with shaded areas indicating standard deviation. The realism meta-metric is evaluated on the 2% validation split. Freezing the map encoder results in lower score variance and improved simulation realism throughout training.

## D  ETHICS STATEMENT

This work adheres to the ICLR Code of Ethics. No human-subjects or animal experimentation was involved. All datasets used—primarily the Waymo Open Motion Dataset (WOMD)—were accessed and processed in compliance with their respective licenses and usage guidelines, with no attempt at re-identification or extraction of personally identifiable information. We took care to assess and mitigate potential biases in data and evaluation, and found no evidence of discriminatory outcomes within the scope of our experiments. No experiments were conducted that could raise privacy or security concerns. We are committed to transparency and integrity throughout the research process, including clear reporting of methods, metrics.

## E  THE USE OF LARGE LANGUAGE MODELS (LLMS)

We used a large language model (LLM), specifically OpenAI's ChatGPT-5, as a writing assistant to polish and refine the presentation of our paper. The LLM was employed for grammar correction, clarity improvements, style adjustments, and consistency of terminology across sections. Importantly, all technical content, research ideas, model design, experiments, and analyses were conceived, implemented, and validated entirely by the authors. The LLM did not contribute to research ideation, algorithm development, experiment design, or interpretation of results. Its role was limited to improving readability and presentation quality.

## F  REPRODUCIBILITY STATEMENT

We have taken multiple steps to ensure the reproducibility of our work. Our dataset and pre-trained backbone are the same as those provided in the open-source CAT-K repository (`https://github.com/NVlabs/catk/tree/main`). Detailed descriptions of the model architecture and all hyperparameters are included in Section A and in the appendix. We also report the official performance of our method on the public WOSAC leaderboard, which enables transparent comparison against other approaches.

## G    REALISM SCORE VISUALIZATION

In this section, we visualize the ego vehicle's *scene realism* and *interaction realism* scores produced by our decomposed discriminator, for both two successful rollouts and two failure cases.

### G.1    SUCCESSFUL CASES

We present two successful rollouts illustrating both straight-road and intersection driving and intersection. As shown in fig. 7, the realism scores remain close to 0.5 throughout the rollout, and all agents drive without collisions or off-road behavior.

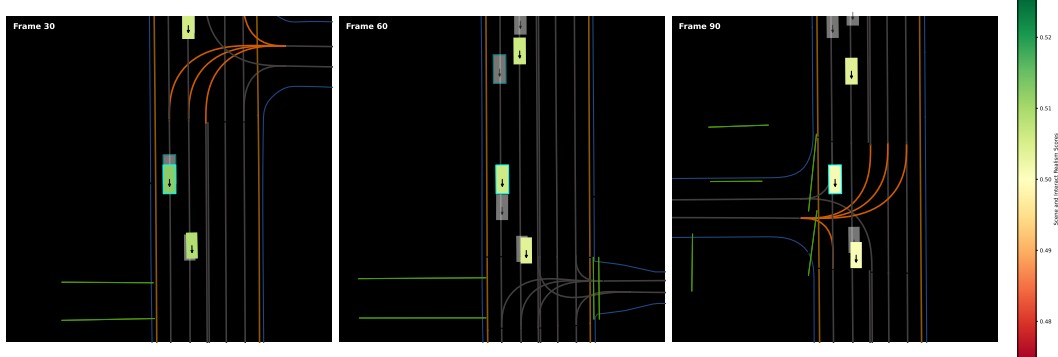

Figure 7: **Straight driving case.** The ego vehicle is shown with cyan edges; its color encodes the *scene realism* score. Surrounding vehicles are colored by their *interaction realism* with respect to the ego. The realism scores remain near 0.5, and all agents drive smoothly without collisions or off-road deviations.

In fig. 8, the simulated ego vehicle chooses a turning maneuver while the ground-truth agent continues straight through the intersection, demonstrating the behavioral diversity captured by our learned traffic model. The slightly lower scene realism score in this case is likely due to low kinematic likelihood.

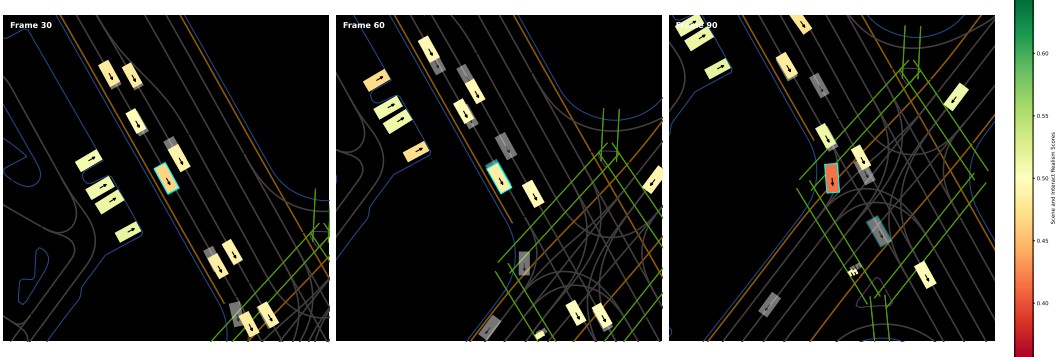

Figure 8: **Intersection case.** The ego vehicle is shown with cyan edges; its color encodes the *scene realism* score. Surrounding vehicles are colored by their *interaction realism* with respect to the ego. The simulated ego vehicle chooses a turning trajectory, whereas the ground-truth agent proceeds straight through the intersection. This illustrates the diversity of behaviors produced by our learned model. The slight decrease in scene realism mainly reflects kinematic deviations rather than unrealistic maneuvers.

## G.2 FAILURE CASES

Figure 9 shows a failure case in which the ego vehicle drifts outside the drivable area. As expected, our discriminator assigns a very low *scene realism* score at those timesteps, correctly indicating the implausible off-road behavior.

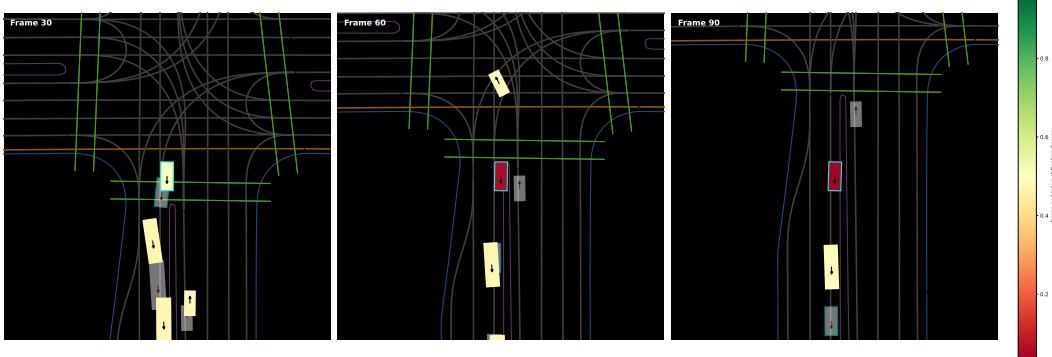

Figure 9: **Off-road failure case.** The ego vehicle is shown with cyan edges; its color encodes the *scene realism* score. Surrounding vehicles are colored by their *interaction realism* with respect to the ego. In the middle and right frames, the ego vehicle violates the road boundary (purple color line) and leaves the drivable surface. Our discriminator correspondingly outputs a very low scene realism score, accurately detecting the off-road behavior.

Figure 10 presents a failure case where the ego vehicle collides with a neighboring agent. Here, our discriminator produces a very low *interaction realism* score, correctly flagging the implausible collision event.

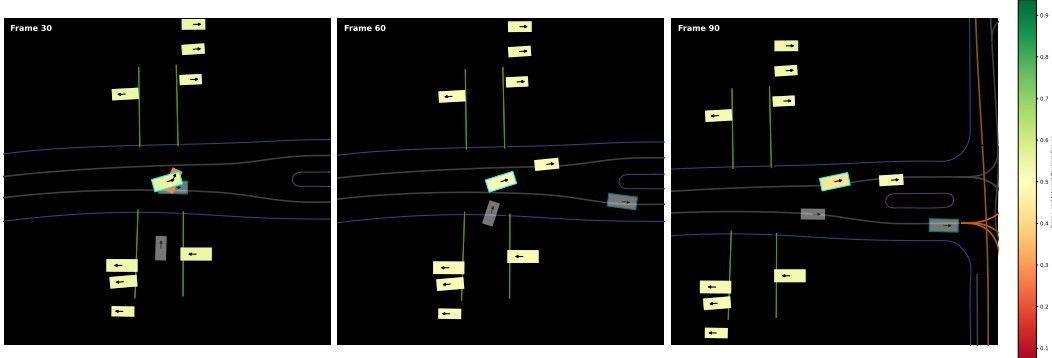

Figure 10: **Collision failure case.** The ego vehicle is shown with cyan edges; its color encodes the *scene realism* score. Surrounding vehicles are colored by their *interaction realism* with respect to the ego. In the left frame, the ego vehicle makes contact with another vehicle. The discriminator assigns a very low interaction realism score to this agent pair, properly reflecting the implausibility of the collision.

