# OpenReview forum: "DecompGAIL: Learning Realistic Traffic Behaviors with Decomposed Multi-Agent Generative Adversarial Imitation Learning"
_ICLR.cc/2026/Conference — ICLR 2026 Poster_

### Official Review · Reviewer_dJoy · 2025-10-26

**Soundness:** 3
**Presentation:** 4
**Contribution:** 3
**Rating:** 8
**Confidence:** 4

**Summary:**

* This paper addresses the problem of training instability in multi-agent GAIL for realistic traffic simulation.
* It identifies irrelevant interaction misguidance as a key issue, where the discriminator incorrectly penalizes a realistic ego agent because of unrealistic interactions among its neighbors.
* The proposed solution DecompGAIL introduces a decomposed discriminator architecture that explicitly separates realism into (1) ego-map realism and (2) pairwise ego-neighbor realism.
* This decomposed design structurally filters out the misleading neighbor-neighbor and neighbor-map interaction signals that cause instability.
* The method is enhanced with a social PPO objective that augments the agent’s reward with a distance-weighted sum of its neighbors' rewards, promoting overall population realism.
* The method achieves SOTA on WOSAC.

**Strengths:**

* DecompGAIL achieves state-of-the-art results on WOMD sim agents and demonstrates significantly improved training stability.
* Irrelevant interaction misguidance is a novel problem that the authors discovered. GAIL has been notoriously difficult to train.
* The authors show strong empirical validation that the proposed approach significantly improves training stability and in addition achieves strong results on WOSAC.
* The ablation study validates that each proposed component contributes to the strong results.

**Weaknesses:**

* While the solution of decomposing the discriminator is sound it is engineered and strongly dependent on the input features that are used by the sim agent model. For example, it’s unclear how this approach could be applied to a simulator model that leverages images and sensor sim.

**Questions:**

* Are there failure cases or poor quality examples that could be included?

---

> ### Author Response · Authors · 2025-11-22
> **Official Comment by Authors**
>
> We sincerely thank the reviewer for your encouraging feedback. We are glad that the reviewer found the method effective, well-motivated, and novel, and appreciated the empirical rigor of our experiments and ablations. Below we provide point-by-point responses to all questions.
>
> ---
>
> ### **W1: Dependency on engineered features; unclear applicability to image-based simulators**
>
> We appreciate this concern. Although our experiments use tokenized map and agent representations, the **core idea of decomposition is modality-agnostic**. For image-based or sensor-simulated environments, the same principles apply:
>
> * **Scene realism** can be computed from an input image where only ego and static context (road geometry, map elements, background) is rendered or retained.
> * **Interaction realism** for each ego–neighbor pair can be computed from images where only the ego and one specific neighbor are rendered, either by masked rendering or layered rasterization.
>
> Thus, the decomposed design is compatible with image-based simulators.
>
> ---
>
> ### **Q1: Are there failure cases or poor-quality examples that could be included?**
>
> Thank you for the helpful suggestion. In response, we have added two failure cases in **Appendix F.2**, covering off-road behavior and collision.
>
> In both cases, we visualize the **scene realism** and **interaction realism** predicted by the decomposed discriminator. The discriminator assigns **appropriately low scene realism** for off-road behavior and **low interaction realism** for collision events, illustrating that it produces *meaningful and interpretable signals* in failure scenarios.

---

### Official Review · Reviewer_sEmo · 2025-10-30

**Soundness:** 2
**Presentation:** 2
**Contribution:** 2
**Rating:** 4
**Confidence:** 5

**Summary:**

This work proposes DecompGAIL, which aim to address the instability of the Generative Adversarial imitation Learning. The authors proposes to decomposes the realism to ego-map and ego-neighbor to avoid weakly relevant neighbor–neighbor interactions. Alghough it shows better training stability, the model is built upon a very strong pretrained backbone, and the results are saturated (only minimal gain) on the overall realism. The authors should conduct a more comprehensive study of why do we need Generative Imitlation Learning framework, and highlight how this GAIL-finetuning provided a different aspects, dimensions of traffic simulation

**Strengths:**

- The paper is well-written and the proposed Decomposed GAIL method is more stable than prior GAIL baselines.
- DecompGAIL achieves competitive results on the Sim Agent Challenge 2025, though the gain is very small.
- The main potential of this work is the discriminator, which can provide a useful realism signal compared to prior metrics (see weakness section)

**Weaknesses:**

- Qualitative results are not interesting and are very similar to previous works
- Overall, DecompGAIL’s advantage compared to prior works is unclear, given that the performance improvements are very small (± 0.01 realism score).
- The pretrained backbone already attains a high discriminator score (~0.5) from Figure 3, which suggests the added GAIL module provides minimal gain.
- I suggest the authors to start with a underperformed backbone w/ GAIL finetuning, and the main potential of this work is to  showcase the discriminator:whether it provides a uesful realism signal, compared to prior metrics such as Waymo Sim Agents Challenge
- In Sec 5.4, the authors claimed that BC suffers from a covariate shift problem by mentioning higher collision likelihood and very minimal gain on realism. This may not hold true as the Sim Agent Challenge measures distributional realsim instead of rule satisfaction; see [1].


[1] Wang M., Wang J., Ye T., Chen J., Yu K. (2025). Do LLM Modules Generalize? A Study on Motion Generation for Autonomous Driving. CoRL.

**Questions:**

- How does this work compared to supervised fine-tuning, what are the advantages of using GAIL, given that the gain is very minimal  compared to the original SMART 0.05
- The definition and handling of “ego” vs “neighbor” agents is unclear: during training, are all agents’ policies updated simultaneously, or is only the ego agent’s policy learned while neighbors remain fixed? How does this affect memory usage and training dynamics (especially when using a decomposed discriminator)?
- Please address the weakness sections

---

> ### Author Response · Authors · 2025-11-22
> **Official Comment by Authors**
>
> We thank the reviewer for the constructive feedback. Below we address each concern and provide additional empirical evidence and clarifications.
>
> ---
> ### **W1: Qualitative results are not interesting and are very similar to previous works**
>
> Thank you for the helpful comment. Our intention was not to emphasize visually complex scenes as in previous works, but to illustrate that the *decomposed discriminator produces realism-consistent signals*. To address your concern, we have added additional qualitative examples in **Appendix F**, including both **successful** and **failure** cases. These visualizations explicitly show the **scene** and **interaction** realism scores produced by our discriminator, highlighting how the two components respond appropriately to off-road behavior, near-collisions, and realistic interactions—offering qualitative insights beyond previous works.
>
> ---
>
> ### **W2: Overall, DecompGAIL’s advantage is unclear, given that improvements are small (≈ 0.01 meta-metric)**
>
> Our primary contribution is *not* a new leaderboard-topping model, but a general solution to **instability in multi-agent GAIL**, via the identification of the **irrelevant interaction misguidance** problem and the introduction of **DecompGAIL** as a general framework.
>
> Regarding performance: although the absolute improvement (~0.005–0.01) seems numerically small, it is considered **significant** within the WOSAC setting due to:
>
> * extremely tight metric scales,
> * saturation among strong SMART-based models,
> * the meta-metric aggregating >10 likelihood terms,
> * leaderboard variations as small as ±0.003 commonly being treated as meaningful.
>
> More importantly, **DecompGAIL consistently outperforms all prior fine-tuning methods built on the same SMART-tiny backbone** on test set, as shown below:
>
> | Method                | Metametric | Δ vs Baseline | Δ Ratio to Oracle Gap|
> | --------------------- | ---------- | ------------- | ----------------- |
> | Logged Oracle (expert data)| 0.8194     | +0.0380       | 100%              |
> | **DecompGAIL**        | **0.7864** | **+0.0050**   | **13.16%**        |
> | SMART-R1 (CLSFT+RL)   | 0.7858     | +0.0044       | 11.58%            |
> | RLFTSim (RL)          | 0.7857     | +0.0043       | 11.32%            |
> | CLSFT (SFT)           | 0.7846     | +0.0032       | 8.42%             |
> | SMART-tiny (baseline) | 0.7814     | +0.0000       | 0.00%             |
>
> These results demonstrate that DecompGAIL provides the strongest gain among all existing fine-tuning baselines, despite operating under the same policy model structure.
>
> ---
>
> ### **W3: Pretrained backbone already attains a ~0.5 discriminator score—does GAIL add benefit?**
>
> We apologize for the confusion. The policy is pretrained, but the **discriminator is trained from scratch** during DecompGAIL finetuning. A score near 0.5 at begining simply reflects an *untrained* discriminator that cannot yet distinguish expert from policy samples.
>
> After training begins:
>
> * the discriminator learns meaningful distinctions,
> * naive PS-GAIL becomes unstable,
> * DecompGAIL produces stable ~0.5 mean with low variance, as expected for a converged GAN-style discriminator.
>
> Thus, the initial ~0.5 value does not indicate that GAIL provides little benefit—only that the discriminator begins untrained.
>
> ---
>
>
> ### **W4: Start with an underperforming backbone to showcase discriminator effectiveness**
>
> Thank you for this excellent suggestion. We conducted new experiments using two deliberately weakened backbones:
>
> 1. **SMART-micro** (0.66M parameters; half hidden dimension; 1 attention layer)
> 2. **SMART-tiny-3ep** (trained with only 10% of standard BC training time—3 epochs)
>
> The results on the validation subset show *substantial improvements* from DecompGAIL:
>
> | Method                          | Meta-metric | Δ vs Baseline |Δ Ratio to Oracle Gap |
> | ------------------------------- | ----------- | ------------- | ------------------- |
> | Logged Oracle (expert data)      | 0.8235      | —             | 100%                |
> | SMART-micro                     | 0.7683      | 0.0000        | 0.0%                |
> | **SMART-micro + DecompGAIL**    | **0.7817**  | **+0.0134**   | **35.3%**           |
> | SMART-tiny-3ep                  | 0.7590      | 0.0000        | 0.0%                |
> | **SMART-tiny-3ep + DecompGAIL** | **0.7885**  | **+0.0295**   | **77.6%**           |
>
> These results reinforce that **DecompGAIL provides strong improvements even on weak backbones**, validating that our contribution is not merely pushing a saturated model but solving a fundamental training issue.
>
> ---

---

> ### Author Response · Authors · 2025-11-22
> **Official Comment by Authors**
>
> ### **W5: Covariate shift argument may not apply since SimAgent measures distribution realism**
>
> Thank you for raising this important point. We agree that the Sim Agents Challenge evaluates **distributional realism**, not rule satisfaction. Our intention was not to treat collision likelihood as a rule-based metric, but rather as a **distribution-matching measure**, fully consistent with the WOSAC definition.
>
> To clarify:
>
> * Collisions **do appear** in the logged Waymo dataset due to sensing noise and annotation artifacts. Consequently, a policy that never collides may obtain a *worse* collision likelihood if it fails to reproduce the expert distribution of collision events. Thus, the metric reflects **distribution alignment**, not whether the agent obeys safety rules.
>
> * The observation that BC exhibits a *lower collision likelihood* refers to **covariate shift in distribution space**. As BC drifts off the expert manifold in closed-loop rollouts, its induced state–action distribution deviates from the logged data. This mismatch leads to:
>
>   * entering states where the expert almost never collides → reducing likelihood alignment, and
>   * failing to reproduce subtle collision events present in the expert data → further lowering the likelihood.
>
> In other words, the effect is a consequence of **distribution mismatch caused by covariate shift**, not a claim about BC violating safety rules.
>
>
> ---
>
> ### **Q1: Why use GAIL instead of supervised fine-tuning?**
>
> Empirically, DecompGAIL provides stronger improvement than CLSFT (+13.16% vs. +8.42% of the oracle gap) on test set:
>
> | Method                      | Metametric      | Δ vs Baseline (0.7814) | Δ Ratio to Oracle Gap |
> | --------------------------- | ---------- | ---------------------- | ------------------- |
> | Logged Oracle (expert data) | 0.8194     | +0.0380                | 100%                |
> | **DecompGAIL**              | **0.7864** | **+0.0050**            | **13.16%**          |
> | SMART-tiny-CLSFT (CLSFT)       | 0.7846     | +0.0032             | 8.42%              |
>
>
> Theotherically, SFT has following drawbacks compared with GAIL-based method:
> * SFT does not expose the policy to its own rollout distribution (**only top-k policy distribution**); covariate shift persists. But the GAIL fully expose the policy to its own rollout distribution.
> * SFT “recovers” by forcing the model back to the expert future trajectory, even when the model’s prior steps deviate from expert behavior. This can cause state-action distribution mismatch. Instead, GAIL provides **state-distribution-aligned corrections** by guiding the policy to the expert state-action distribution.
>
> ---
>
>
>
> ### **Q2: Clarification of “ego” vs. “neighbor” agents—do all policies update simultaneously?**
>
> Thank you for pointing out the ambiguity. We clarify:
>
> * During training, **every agent is treated as the ego** when computing its own reward.
> * After computing all agents’ rewards, **all agents update their shared policy simultaneously using PPO** (parameter sharing).
>
> Memory and compute considerations:
>
> * DecompGAIL is **memory-efficient** because the realism is computed via lightweight MLPs and graph scatter operations. DecompGAIL **reduces memory compared to PS-GAIL**, which removes agent–agent attention layers.
> * Training dynamics are **more stable**, as shown in Fig. 3.

---

> > ### Comment · Reviewer_sEmo · 2025-11-25
> >
> > Thank you for the clarifications and additional experiments. I still think the gains on the main benchmark are small and likely within these models’ variance. However, the improvements on underperforming backbones demonstrate the effectiveness of the proposed GAIL finetuning. I will update my score accordingly.

---

### Official Review · Reviewer_bk4B · 2025-10-31

**Soundness:** 3
**Presentation:** 3
**Contribution:** 3
**Rating:** 4
**Confidence:** 4

**Summary:**

This paper addresses a key limitation in multi-agent Generative Adversarial Imitation Learning (GAIL) for traffic simulation, where existing discriminators penalize realistic agents simply because other nearby agents behave unrealistically. The authors call this phenomenon irrelevant interaction misguidance.
To solve this, they propose DecompGAIL, which decomposes the discriminator into two terms: ego–map realism (how the ego vehicle interacts with the static environment) and ego–neighbor realism (how it interacts with relevant nearby agents). Irrelevant neighbor–neighbor or distant-agent interactions are filtered out through distance-weighted aggregation. In addition, a social PPO reward encourages coherent scene-level realism by adding distance-weighted neighbor rewards to each agent’s objective. Experiments on the WOMD Sim Agents 2025 benchmark show that DecompGAIL improves realism metrics and training stability compared with prior multi-agent GAIL baselines.

**Strengths:**

- Clearly identifies and mitigates the irrelevant interaction misguidance problem.

- The decomposition is simple, computationally efficient, and compatible with existing frameworks.

- Demonstrates improved realism and stability on public benchmarks with comprehensive ablations.

- Offers a general recipe that can transfer to other multi-agent imitation settings.

**Weaknesses:**

- Comparisons omit Wasserstein or gradient-penalized discriminators, making it unclear whether decomposition alone drives the gains.

- The social reward could unintentionally amplify correlated neighbor behaviors.

- The related work section omits several closely related papers:

A. Kuefler, J. Morton, T. Wheeler, and M. J. Kochenderfer, “Imitating Driver Behavior with Generative Adversarial Networks,” IEEE Intelligent Vehicles Symposium (IV), 2017, pp. 204–211.

R. P. Bhattacharyya, B. Wulfe, D. J. Phillips, A. Kuefler, J. Morton, R. Senanayake, and M. J. Kochenderfer, “Modeling Human Driving Behavior through Generative Adversarial Imitation Learning,” CoRR, 2020.

H. Chen, T. Ji, S. Liu, and K. Driggs-Campbell, “Combining Model-Based Controllers and Generative Adversarial Imitation Learning for Traffic Simulation,” IEEE ITSC 2022, pp. 1698–1704.

K. Brown, K. Driggs-Campbell, and M. J. Kochenderfer, “Modeling and Prediction of Human Driver Behavior: A Survey,” arXiv:2006.08832, 2020.

**Questions:**

How sensitive is performance to the decay parameters $\alpha$ and $\beta$ for interaction weighting and social rewards?

When freezing or fine-tuning the map encoder during discriminator training, how does stability or realism change?

Could the social reward cause feedback loops where agents learn to exploit mutual rewards without improving realism?

---

> ### Author Response · Authors · 2025-11-22
> **Official Comment by Authors**
>
> We thank the reviewer for the encouraging assessment and for recognizing the novelty, robustness, generalizability, and empirical strength of our proposed **DecompGAIL** framework. Below, we respond to each weakness and question in detail.
>
> ---
>
>
> ### **W1: Comparisons omit Wasserstein or gradient-penalized discriminators.**
>
> Thank you for raising this important point. Our approach is **orthogonal** to discriminator regularization techniques such as **WGAN-GP** and **R1/R2 gradient penalties**.
>
> To evaluate their influence, we added WGAN-GP and R1/R2 penalties (weight = 1) to both **PS-GAIL** and **DecompGAIL**. Key results are shown below (full table in Appendix B):
>
> | **Model**            | **Metametric ↑** | **Kinematic ↑** | **Collision ↑** |
> | -------------------- | ---------------- | --------------- | --------------- |
> | PS-GAIL              | 0.7674           | 0.4830          | 0.9494          |
> | PS-GAIL + WGAN-GP    | 0.7723           | 0.5011          | 0.9538          |
> | PS-GAIL + R1         | 0.7776           | 0.5011          | 0.9704          |
> | PS-GAIL + R2         | 0.7706           | 0.5037          | 0.9484          |
> | DecompGAIL + WGAN-GP | 0.7865           | **0.5186**      | 0.9796          |
> | DecompGAIL + R1      | 0.7861           | 0.5157          | 0.9734          |
> | DecompGAIL + R2      | 0.7840           | 0.5145          | 0.9743          |
> | **DecompGAIL**       | **0.7889**       | 0.5148          | **0.9837**      |
>
> **Findings:**
>
> * Gradient penalties modestly stabilize **PS-GAIL**, but do **not** resolve its core issue: the discriminator is still misled by irrelevant neighbor–neighbor interactions.
> * Adding penalties to **DecompGAIL** slightly improves kinematics but **hurts collision likelihoods** because gradient regularization enforces smoothness, whereas collision signals are inherently **non-smooth** and require sharp decision boundaries.
>
>
> ---
>
> ### **W2: The social reward could unintentionally amplify correlated neighbor behaviors.**
>
> We agree this is an important concern. This is exactly why we adopt a **distance-decayed neighborhood reward**, ensuring influence decreases exponentially with distance. This design captures the fact that interactions between nearby agents are also weakly correlated with the ego’s own behavior.
>
> Our ablations show:
>
> * Using **mean neighbor reward** (Table 2), or
> * Using a **large decay range** β (Figure 4)
>
> both worsen realism because they overweight distant neighbors, amplifying irrelevant neighbor–neighbor interactions. Selecting a **small β** restricts influence to meaningful near-field interactions, avoiding the amplification issue.
>
> ---
>
> ### **W3: Missing related work.**
>
> Thank you for pointing this out. We have updated the **Related Work** section to incorporate all missing references.
>
> ---
>
> ### **Q1: Sensitivity to decay parameters α and β.**
>
> We conducted thorough sweeps over α and β for both **interaction weighting** and **social reward weighting**.
>
> #### **Interaction weight sweep (without social rewards)**
>
> | α \ β | 1       | 2.5     | 5       | 10      |
> | ----- | ------- | ------- | ------- | ------- |
> | 2.5   | 0.78117 | 0.78528 | 0.78353 | 0.78448 |
> | 5     | 0.78157 | 0.78650 | 0.78601 | 0.78191 |
> | 10    | 0.78421 | 0.78700 | 0.78504 | 0.77961 |
> | 20    | 0.78514 | 0.78674 | 0.78473 | 0.77732 |
>
> **Findings:**
>
> 1. The model is **more sensitive to β (interaction range)** than α.
> 2. Extremely small or large β overemphasizes very near or very distant neighbors, degrading realism.
>
> ---
>
> #### **Social reward sweep** (fixed optimal interaction α=10, β=2.5)
>
> | α \ β | 1       | 2.5     | 5       | 10      |
> | ----- | ------- | ------- | ------- | ------- |
> | 0.5   | 0.78630 | 0.78810 | 0.78683 | 0.78692 |
> | 1     | 0.78840 | 0.78888 | 0.78839 | 0.78743 |
> | 2     | 0.78608 | 0.78750 | 0.78711 | 0.78620 |
> | 10    | 0.78296 | 0.78458 | 0.78370 | 0.78040 |
>
> **Findings:**
>
> * Social reward parameters have **weaker effect** than interaction weights.
> * Large α or β make the neighborhood weights nearly uniform → increases noise → reduces realism.
>
> ---
>
> ### **Q2: Freezing vs. fine-tuning the map encoder.**
>
> We added experiments comparing both settings (Appendix C, Fig. 6).
>
> **Result:**
> Freezing the map encoder leads to **slightly better stability** and **marginally higher realism**.
>
> ---
>
> ### **Q3: Could social rewards cause feedback loops?**
>
> We analyzed this potential issue theoretically.
>
> Consider two agents:
> r^s_a = r_a + λ_ab * r_b,
> r^s_b = r_b + λ_ab * r_a,
> where λ_ab > 0.
>
> In this formulation, increasing either social reward necessarily requires increasing r_a or r_b—the realism rewards produced directly by the discriminator. Since (r^s_a) and (r^s_b) depend **not directly on each other**, there is *no* mechanism for a self-reinforcing feedback loop in which agents inflate rewards without improving realism.

---

> > ### Comment · Reviewer_bk4B · 2025-11-26
> >
> > Thank you for the detailed rebuttal. The responses address my concerns, and I have raised my score.

---

### Meta-Review · Area_Chair_BQP5 · 2026-01-10

**Summary:**

This paper introduces DecompGAIL, a decomposed discriminator for multi-agent GAIL that separates ego-map and ego-neighbor realism to mitigate instability caused by irrelevant neighbor interactions. It demonstrates improved stability and state-of-the-art realism on the WOMD Sim Agents benchmark. Reviewers agree the core idea of identifying “irrelevant interaction misguidance” and addressing it through discriminator decomposition is well-motivated. Initial concerns about missing comparisons, social-reward feedback, hyperparameter sensitivity and backbone saturation were largely addressed through added experiments and clarifications. Remaining concerns are on the modest absolute performance gains and whether the contribution is primarily methodological rather than empirical. Given that major technical concerns were resolved and the method provides a solution to a known instability, it is recommended for acceptance to ICLR. It is suggested that the authors incorporate all reviewer suggestions in the final version of the paper.

**Reviewer Concerns:**

### Addressed concerns
* **bk4B:** Comparisons omit Wasserstein/gradient-penalized discriminators. The author response added WGAN-GP, R1 and R2 to both PS-GAIL and DecompGAIL, showed decomposition still outperforms regularization alone and penalties may hurt collision realism.
* **bk4B:** Social reward may amplify correlated neighbor behavior or induce feedback loops. The author response justified distance-decayed weighting, showed with ablations and analysis that large beta or uniform weighting degrades realism and that rewards cannot self-reinforce without increasing discriminator-based realism.
* **sEmo:** Illustrate advantage with underperforming backbone. The author response provided oracle-gap analysis and new experiments on weakened backbones (SMART-micro, undertrained models) showing substantially larger gains.
* **sEmo:** Qualitative results are uninformative about the discriminator’s added value. The author response adds more results to show interpretability beyond metrics.
* **sEmo:** Ambiguity about ego or neighbor training and whether GAIL adds benefit over supervised fine-tuning. The author response clarifies that all agents are treated as ego with shared policy updates and provided quantitative comparison to supervised fine-tuning.
* **djoy:** Dependence on input features. The author response clarifies that the core idea of decomposition is applicable across modalities.

### Unaddressed concerns
* **sEmo:** Small improvements. The author response discusses significance in the WOSAC setting, but concern still remains that improvements are within error margin.

**Reviewer Scores:**

* **bk4B:** Initially rated the paper 4, but the rebuttal addresses all concerns and would likely raise their score to 6 or 8.
* **sEmo:** Initially rated the paper 4, but the rebuttal addresses most concerns and would likely raise their score to 6.
* **djoy:** Initially rated the paper 8, the rebuttal addresses all concerns and would likely maintain their score at 8.

---

### Decision · Program_Chairs · 2026-01-26

Accept (Poster)